# Younger Americans are less politically polarized than older Americans about climate policies (but not about other policy domains)

**Joshua F. Inwald** [ORCID][1]*, **Wändi Bruine de Bruin**[2], **Christopher D. Petsko**[3]

1 Department of Psychology, University of Southern California, Los Angeles, California, United States of America, 2 Sol Price School of Public Policy and Department of Psychology, University of Southern California, Los Angeles, California, United States of America, 3 Kenan-Flagler Business School, University of North Carolina at Chapel Hill, Chapel Hill, North Carolina, United States of America

* inwald@usc.edu

**Data Availability Statement:** All raw data files are available for free download from the American National Election Studies: https://electionstudies.org/data-center/.

## Abstract

Political polarization of Americans' support for climate policies often impedes the adoption of new, urgently needed climate solutions. However, recent polls suggest that younger conservatives favor adopting pro-climate policies to a greater degree than older conservatives, resulting in less political polarization among younger Americans relative to older Americans. To better understand these patterns, we analyzed Americans' support for various climate policies from 1982–2020, across 16 waves of historical, nationally representative survey data from the American National Election Studies (total $N = 29,467$). Regression models consistently show that, since 2012, younger Americans have been less politically polarized than older Americans on support for climate policies. Before 2012 and on non-climate policy topics, we did not find consistent statistical evidence for political polarization varying with age. These findings can inform policy debates about climate change and offer hope to environmentalists and policymakers who seek to build broad consensus for climate action at the policy level.

## 1. Introduction

Global climate change poses immense threats to human civilization and natural environments [1]. Consequently, organizations such as the Intergovernmental Panel on Climate Change (IPCC) and World Health Organization call for rapid adoption of climate change mitigation and adaptation measures (henceforth, "climate policies") [2, 3].

While these calls have resulted in international climate treaties like the 2015 Paris Agreement, the United States has been slow to implement policies that would reduce the pace of climate change [4]. Indeed, the United States is projected to underachieve its 2030 greenhouse gas emissions reduction goals by 22%–45%, even after accounting for 2022's bipartisan, climate-centric Inflation Reduction Act [5]. Without additional concerted action by large greenhouse-gas emitters—including the United States—there is concern that it will be nearly

**Funding:** The author(s) received no specific funding for this work.

**Competing interests:** The authors have declared that no competing interests exist.

impossible to reach the Paris Agreement's goal of limiting global temperature increases to 2°C compared to preindustrial averages [6, 7].

One pathway towards meeting climate goals involves leveraging the public's heightened concern for climate change. In the 2019 World Risk Poll, a majority of people in the United States and around the world reported being concerned about climate change [8]. Democratic governments are generally responsive to their constituents' opinions on environmental issues, with higher levels of climate change concern and issue salience both positively linked with government adoption of climate policies [9–11].

## 1.1 Political polarization of Americans' support for climate policies

Unfortunately, in the United States, people's attitudes about climate policies have been polarized along political lines [12–14]. Historically, political liberals in the United States have been more concerned about climate change and expressed greater demand for policy solutions, compared to conservatives [15]. Even in the last five years, liberals have been much more likely to agree that there is anthropogenic climate change [16], to self-report engaging in pro-environmental behaviors [17], and to say the national government should prioritize "dealing with climate change and global warming" [18]. Researchers have concluded that such polarization constitutes a barrier towards the adoption of climate policies in the United States, at both national and state levels [19–21].

One explanation for Americans' polarization on climate policies is provided by cultural cognition theory, which argues that opposing beliefs on the values that underpin an ideal society lead to political differences in climate change concerns and policy preferences [22]. Under this account, political liberals are motivated by communitarian worldviews and political conservatives by individualistic worldviews [22, 23]. As a result, liberals are more likely to support government interventions on climate because they perceive climate change as a significant threat to collective welfare; conversely, conservatives are more likely to question the merits of climate policies that might infringe on personal autonomy [23]. Americans also tend to trust media sources that are "culturally congruent", or in alignment, with their political ideology, leading to enhanced polarization [24]. Cultural cognition theory has also been proposed as an explanation for Americans' politically polarized attitudes about other threats to public wellbeing, such as nuclear weapons [25].

Political polarization may also persist because of motivated reasoning, or the tendency to selectively interpret evidence that confirms preexisting beliefs [26]. For example, one nationally representative survey of Americans in 2009 found that, for liberals, greater knowledge about climate change was associated with heightened support for climate mitigation policies, whereas the opposite association was found in conservatives [27]. The evidence base for scientific literacy leading to motivated reasoning about climate policy is mixed, however [26], including two nationally representative U.S. studies reporting null effects for motivated reasoning as a driver of climate policy polarization [28, 29].

## 1.2 Generational differences in climate policy attitudes and decision-making over time

Irrespective of the roots of politically polarized views on climate change, the strong association between Americans' political ideology and their support for climate policies might not hold among young Americans today [30]. Nationally representative surveys from 2020 and 2021 found that younger Republicans (Millennials and Generation Z) were more supportive of climate policies and are more concerned about climate change compared to older Republicans

(Baby Boomers and older). In contrast, Democrats of all ages were highly supportive of climate policies [31, 32].

However, it is unclear whether younger adults in the United States have *always* been less politically polarized in their support for climate policies. Existing longitudinal studies on the evolution of Americans' climate policy opinions over time do not directly address this question. For example, existing analyses are limited by a reliance on regional survey data rather than national samples [33], or by collapsing across years of public opinion data without analyzing time-series trends [13]. Additionally, such studies do not compare how patterns of support for climate policies compare to public support for other policy domains [13, 33]. Both studies do show that younger Republicans have been more supportive of climate policies than older Republicans since 2016.

Why are younger Republicans today more supportive of climate policies than older Republicans? One potential explanation is due to cohort differences between younger versus older conservatives. Younger conservatives are part of a generation that is more aware of and more concerned about climate change [34, 35]. Additionally, today's news media prominently discusses climate change policies and affords high social capital to many climate activists, especially on social media [36, 37]. Younger conservatives may therefore be more influenced by social norms to address climate change [38]. Moreover, younger conservatives may be more concerned about climate change simply because they are more likely to be alive when more serious consequences of climate change manifest [39].

Additionally, older adults might be more likely than younger adults to rely on their political ideology when responding to climate policy survey questions. Two findings from the literature on aging and decision-making offer mechanisms for this hypothesis. First, socioemotional selectivity theory posits that older people are highly motivated to maintain positive emotional wellbeing, compared to younger adults [40]. Accordingly, if negative emotions like fear come to mind when being asked to think about climate change policies, older adults might be more motivated to quickly resolve upsetting thoughts (known as the "positivity effect") [41, 42]. A second finding is that fluid intelligence, or the ability to reason flexibly and solve novel problems, declines as people age [43]. On complex tasks, such as forming opinions on the nuanced topic of climate science, this could result in older people preferentially relying on their political party's views rather than engaging in more deliberate decision-making [44, 45]. Taken together, younger conservatives might be more likely to deeply reflect on climate policy questions and offer answers that conflict with their political ideology.

## 1.3 The current study

The primary goal of this study is to better understand the precise interplay between political ideology and age in predicting support for climate policies. Such knowledge is valuable, as it would aid social scientists in designing psychological interventions and communication strategies that overcome partisan divides on climate policy [46]. Additionally, this manuscript helps contextualize previous research on climate policy polarization in three ways. First, we analyze attitudes towards a diverse set of climate policies. In doing so, we answer recent calls to study public opinion of a wide range of environmental solutions [47]. Second, in analyzing four decades of time series data, we identify when in history current patterns of support for climate policies might have emerged [48]. Third, we compare attitudes about climate change policy to attitudes about other polarized policy domains. Thus, this analysis can speak to the issue of whether differential political polarization based on age is unique to climate change, or whether it instead applies to Americans' policy attitudes more broadly.

In a secondary analysis of nationally representative surveys conducted from 1982–2020 by American National Election Studies (ANES; 16 waves, ranging from $n = 822$ to $n = 6{,}531$, combined sample $N = 29{,}467$), this paper sought to answer the following research questions:

Research Question 1: Are younger Americans today less politically polarized than older Americans when it comes to support for climate policies?

Research Question 2: If younger Americans are less politically polarized than older Americans in support for climate policies, when in time did this pattern first emerge?

Research Question 3: Are younger Americans less politically polarized than older Americans when it comes to policy preferences in non-climate policy domains?

## 2. Materials & methods

### 2.1 American National Election Studies: Overview

Since 1948, the American National Election Studies (ANES) has conducted public opinion surveys. Each ANES survey recruits a fresh cross-sectional panel of Americans, selected for national representativeness using an area probability sampling frame. All ANES data are freely available for download: https://electionstudies.org/data-center/.

### 2.2 American National Election Studies: Participants

ANES interviewed Americans aged 18 and older, with a lower age limit (15 or 16) in earlier years. All participants provided informed consent. Respondents could select an English- or Spanish-language interview. In 2004 and in all prior waves, ANES surveys were administered face-to-face or over telephone. Since 2008, ANES has also allowed for self-administered online surveys. In 2020, ANES used online video interviews to safeguard against COVID-19. ANES compensated participants for their time. Before 2004, ANES conducted this survey every two years, but since 2004, only every four years. ANES conducts interviews before national elections (usually two months before election day in early November) and attempts a post-election follow-up.

In this paper we analyze all ANES surveys from 1982–2020 (16 waves, combined $N = 29{,}467$). We start with 1982 because this is the first year that ANES began consistently polling Americans on their environmental policy attitudes. Response rates and survey lengths varied year-to-year. A yearly breakdown of ANES sample sizes across political ideology and age groups is available in Supporting Information, see S1 Table.

### 2.3 American National Election Studies: Survey questions

The primary measures used in the study are listed in Table 1. Table 1 shows the survey questions as phrased in ANES 2020 and lists the transformed response scales we used for modeling (e.g. after dichotomizing and reverse-scoring questions as necessary). Question wording sometimes varied year-to-year. Full details on year-to year question design, control variables, and other dependent measures are available in S1 Appendix.

In addition to the repeatedly asked climate policy questions presented in Table 1, ANES also asked survey questions about specific policy preferences in just one or two survey waves over 1982–2020. Before 2008, ANES survey questions tended to focus on topics like pollution and cleaning up nature, as well as one question on global warming in 1996. From 2008, ANES questions focused on climate change-specific policies, such as emissions standards and rising temperatures, reflecting greater public awareness of climate change as a specific threat to the environment. We analyzed all available climate and environmental policy questions to answer

**Table 1. Primary American National Election Studies survey questions.**

| Variable Name | Lowest Scale Point | Highest Scale Point | 2020 Survey Question Wording |
|---|---|---|---|
| **Independent Variables** | | | |
| Political Ideology | 1 = *Extremely liberal* | 7 = *Extremely conservative* | "Where would you place yourself on this scale, or haven't you thought much about this? 1 = extremely liberal, 4 = moderate; middle-of-the-road, 7 = extremely conservative." |
| Age | 15 | 99 | [Age calculated from participants' birth year, month and day] |
| **Climate Policy Time Series Dependent Variables** | | | |
| Federal Spending on the Environment | 0 = *Decrease spending* or *Keep spending the same* | 1 = *Increase spending* | "What about protecting the environment–should federal spending on protecting the environment be increased, decreased, or kept the same?" |
| Environmental Regulations versus Business Interests Trade-off | 1 = *Regulations to protect environment already too much a burden on business* | 7 = *Tougher regulations on business needed to protect environment* | "Some people think we need much tougher government regulations on business in order to protect the environment. Suppose these people are at one end of a scale, at point 1. Others think that current regulations to protect the environment are already too much of a burden on business. Suppose these people are at the other end, at point 7. And, of course, some other people have opinions somewhere in between, at points 2,3,4,5, or 6. Where would you place yourself on this scale, or haven't you thought much about this?"* |
| **Non-climate Time Series Dependent Variables** | | | |
| Federal Spending on Defense | 1 = *Greatly increase defense spending* | 7 = *Greatly decrease defense spending* | "Some people believe that we should spend much less money for defense. Suppose these people are at one end of a scale, at point 1. Others feel that defense spending should be greatly increased. Suppose these people are at the other end, at point 7. And, of course, some other people have opinions somewhere in between, at points 2, 3, 4, 5 or 6. Where would you place yourself on this scale, or haven't you thought much about this?"* |
| Private versus Public Health Insurance | 1 = *Strong preference for private health insurance* | 7 = *Strong preference for government health insurance* | "There is much concern about the rapid rise in medical and hospital costs. Some people feel there should be a government insurance plan which would cover all medical and hospital expenses for everyone. Suppose these people are at one end of a scale, at point 1. Others feel that all medical expenses should be paid by individuals through private insurance plans like Blue Cross or other company paid plans. Suppose these people are at the other end, at point 7. And, of course, some other people have opinions somewhere in between, at points 2, 3, 4, 5, or 6. Where would you place yourself on this scale, or haven't you thought much about this?"* |
| Federal Spending on Welfare Programs | 0 = *Keep welfare spending the same* or *Decrease welfare spending* | 1 = *Increase welfare spending* | "Should federal spending on welfare programs be increased, decreased, or kept the same?" |

Note: Asterisks (*) indicate a question that was reverse-scored from the original response options.

Research Question 2. Full details on these questions' designs are available in S1 Appendix and S2 Table.

## 2.4 Dependent variable coding

For dependent variables with three response options (e.g. *increase*, *decrease* or *keep the same*), we dichotomized responses to allow for easier comparisons across time and question types. Response options to the federal environmental spending question varied year-to-year, so we also dichotomized this question to allow for direct comparisons across survey waves. In dichotomizing, responses were coded such that *1* represented favoring greater environmental intervention, whereas *0* represented opposing environmental regulation or preference for the status quo.

For variables with response scales (mostly *1–7*), we kept the original scale, but reverse-scored many questions to allow for easier comparisons to the binary-transformed variables. We coded those such that the high scale point represented favoring greater environmental intervention and the low scale point represented opposing those intervention or preference for the status quo. The defense spending and private vs public health insurance questions were also reverse-scored to align liberal vs conservative scale direction with the welfare spending question (for which the transformed high scale point indicates the more typical position of political liberals in the United States).

To answer Research Question 2, we created indices of climate policy questions asked in each ANES survey wave, resulting in eight "climate policy support" indices for 1990, 1992, 1996, 2000, 2008, 2012, 2016 and 2020. Climate policy support indices were calculated by averaging z-standardized responses to the constituent questions.

## 2.5 Analytical strategy

Participants' responses to survey questions were modeled using either logistic regression (with logit link function, for binary-coded questions) or multiple linear regression (for Likert-scale questions and the policy indices), using political ideology, age, and their interaction as predictor variables. We used the same set of predictors for all models and analyzed standardized regression coefficients across question types. Models in which the political ideology × age interaction $p$-value was less than $\alpha = 0.05$ were deemed to have statistically significant differential age-based political polarization.

To answer Research Question 1, we examined whether political ideology interacted with age for three climate policy questions asked in ANES 2020: 1) whether federal government spending on protecting the environment should be increased or decreased; 2) whether to prioritize environmental regulations versus business interests; and 3) whether to increase regulations on businesses that emit greenhouse gases.

To answer Research Question 2, we tested for a significant political ideology × age interaction in two analyses of historical ANES environment and climate policy questions. First, in eight ANES survey waves that asked multiple climate policy questions, we averaged normalized responses to all climate policy items available (see S2 Table) to form a within-year "climate policy support index", then regressed this index onto the same model used in the preceding analysis. Second, ANES repeatedly asked two questions over 1982–2020; 1) whether federal government spending on protecting the environment should be increased, and 2) whether to prioritize environmental regulations versus business interests. We analyzed the time series for these items, again testing for a political ideology × age interaction at the 0.05-signifance level. ANES used three different wordings of the environmental regulations versus business interest question across survey waves, but we decided to treat different versions of this question as a unified time-series because they assess the same underlying policy preference (year-to-year question wording is discussed in S1 Appendix).

To answer Research Question 3, we analyzed time-series questions that asked about non-climate policies: 1) defense spending, 2) public versus private health insurance, and 3) welfare spending. We selected these three survey questions because ANES has mostly complete time-series over 1982–2020 and because public opinion data reveals significant political polarization on all three topics [49]. Copying the analytical strategy used in Research Question 2, we tested for a significant political ideology × age interaction in predicting responses to a z-standardized average index of all non-climate questions within each survey wave, as well as testing for the same interaction in all questions analyzed individually.

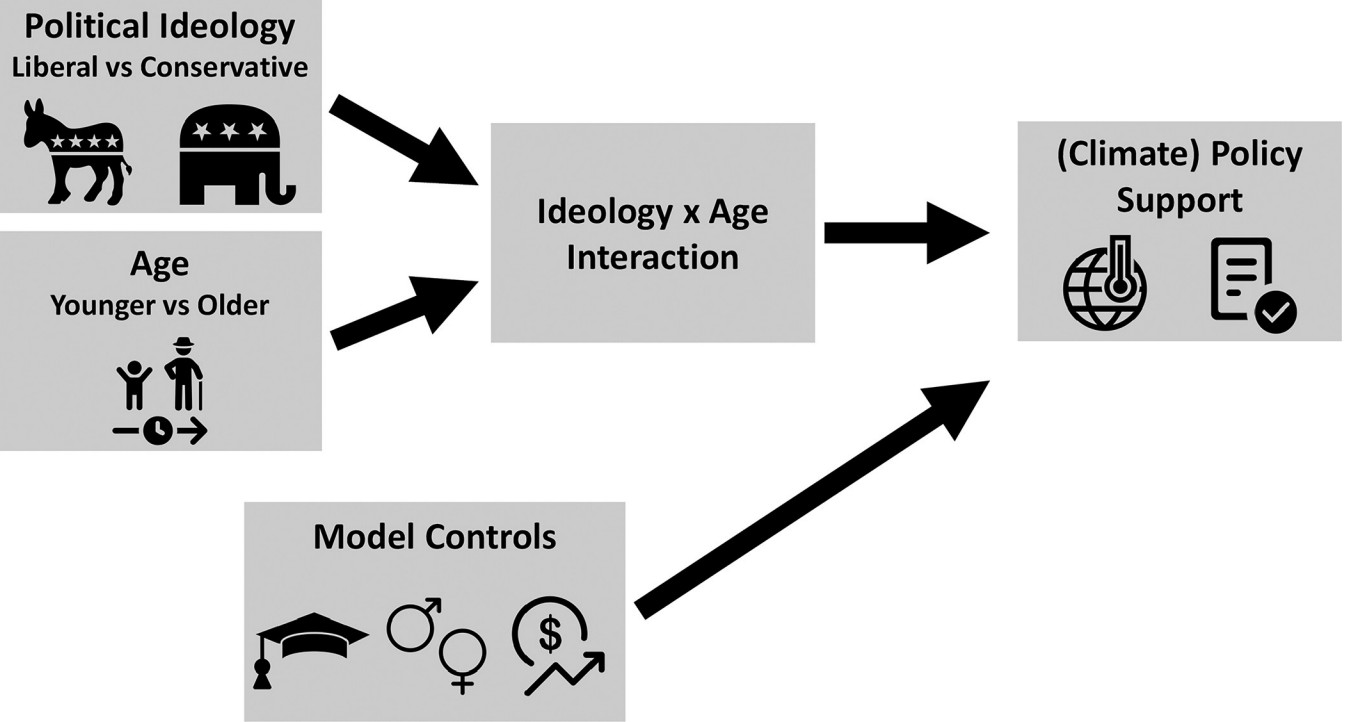

**Fig 1. Visualization of our conceptual model.** Note: Political ideology, age and their interaction are the main independent variables of interest in predicting Americans support for climate policies and other policy topics.

All models controlled for participants' highest level of education completed (dichotomized in our analyses as *College degree* or *No college degree*), the interaction between education and political ideology, gender and household income in the year prior to the survey date, based on association between these demographic variables and climate policy preferences in other studies [50]. We included the interaction between education and political orientation because it significantly improved model fit in 2020 data (which was analyzed first), and because other authors have reported effects of this interaction on support for climate policies [51]. The three-way interaction between political ideology, age and education did not significantly improve model fit, and was thus excluded from our models. Fig 1 provides a visualization of our modeling approach.

We did not use survey weights in our analyses, for three reasons. First, survey weights might be inaccurate due to high proportions of missing data in every survey wave (see S1 Table) [52]. Second, ANES did not calculate survey weights before 1992, so unweighted data analysis offers better consistency over time. Third, there are concerns that using survey weights for regression analysis may lead to less reliable results when analyzing relationships between individual-level variables [53].

Data were analyzed in R version 4.2.2, primarily using functions from the "*stats*" package to model survey responses [54] and functions from the "*effectsize*" package to calculate standardized regression coefficients [55].

## 3. Results

### 3.1 Research finding 1: Younger Americans were less politically polarized than older Americans in 2020

In survey data from the 2020 ANES ($n$ = 6,531), younger Americans were significantly less politically polarized than older Americans in support for three different climate policies. As

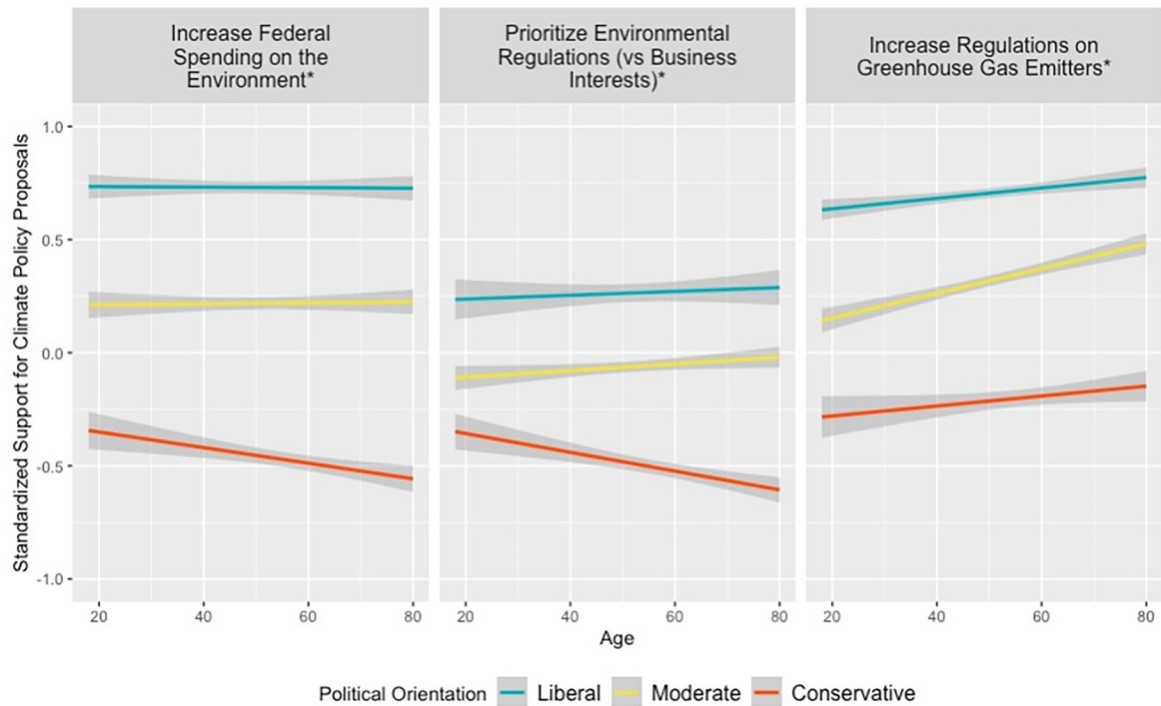

**Fig 2. Younger Americans were less politically polarized than older Americans on support for three climate policies in 2020.** Note: Response variables are z-standardized and displayed with 95%-confidence interval smoothing. Asterisks indicate significant political ideology × age interaction at $p < 0.05$. Full regression outputs are available in S3–S5 Tables. Participants expressed their support for three climate policies, as well as their political ideology (1 = *extremely liberal* to 7 = *extremely conservative*) and age. Models treated political ideology as a continuous variable, but for visualization purposes, people who responded 1 or 2 are coded as "liberal", 3 or 4 or 5 are coded as "moderate", and people who responded 6 or 7 coded as "conservative". For the federal spending on the environment question, participants were asked, "What about protecting the environment–should federal spending on protecting the environment be increased, decreased, or kept the same?", with response coding 1 = *Increase*, 0 = all other responses. For the environmental regulations versus business interests question, participants were asked, "Where would you place yourself on this scale, or haven't you thought much about this? 1 = *Tougher regulations on business needed to protect environment* to 7 = *Regulations to protect environment already too much a burden on business*", though responses were reverse-coded. For the greenhouse gas regulations question, participants were asked, "Do you favor, oppose, or neither favor nor oppose increased government regulation on businesses that produce a great deal of greenhouse emissions linked to climate change?", with responses ranging from 1 = *Oppose regulations a great deal* to 7 = *Favor regulations a great deal*.

can be seen in Fig 2, liberals were highly supportive of these climate policies regardless of age, whereas conservatives' support tended to decline with age, with political moderates' support falling between these groups.

For all three climate policy questions, differential political polarization by age is reflected in the statistically significant standardized regression coefficient and consistent direction for the interaction between political ideology and age (all $p$s < 0.001). For the federal spending on the environment question, the standardized coefficient was $\beta = –0.13$; for the environmental regulations versus business interests question, $\beta = –0.05$; for the question about increasing regulations on greenhouse gas emitters, $\beta = –0.05$. For all three climate change questions, younger Americans' responses were less politically polarized than older Americans. Comprehensive modeling results are available in S3–S5 Tables.

### 3.2 Research finding 2: Younger Americans became less politically polarized than older Americans on climate policies during 2008–2012

Fig 3 shows that, since 2012 but usually not before, younger Americans have been less politically polarized than older Americans in terms of their support for climate policies (full question list in

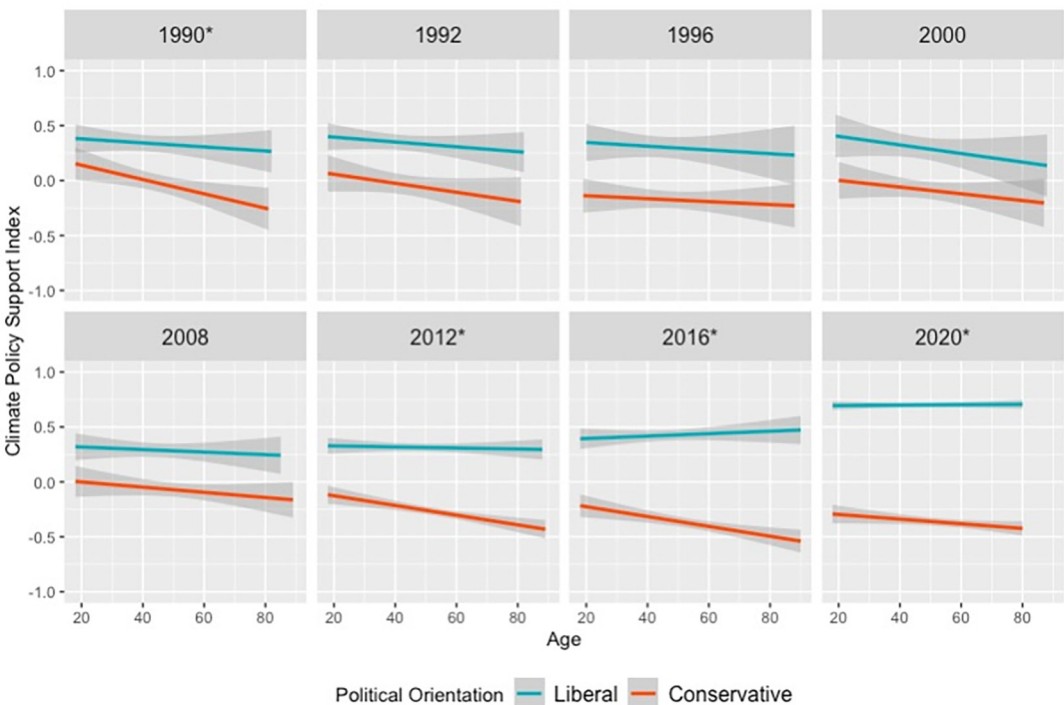

**Fig 3. Younger Americans have been less politically polarized than older Americans on climate policy support since 2012, but not before (Except 1990).** Note: Response variables are displayed with 95%-confidence interval smoothing. Years with an asterisk (e.g. 2012*) indicate significant political ideology × age interaction at $p < 0.05$, whereas non-asterisked survey years do not exhibit significant age-based political polarization. In all surveys, participants expressed their support for climate policies, as well as their political ideology (1 = *extremely liberal* to 7 = *extremely conservative*) and age. Models treated political ideology as a continuous variable, but for visualization purposes, people who responded 1 or 2 are coded as "liberal", and people who responded 6 or 7 coded as "conservative". A version of this chart with political moderates included is available in S1 Fig, and regression models for each year are available in S6–S13 Tables.

S2 Table). We did not detect a significant political ideology × age interaction in 1992, 1996, 2000, and 2008 ($ps > 0.10$), whereas younger Americans were significantly less politically polarized than older Americans in climate policy support in 2012, 2016, and 2020 ($ps < 0.001$). Surprisingly, 1990 also exhibited a significant political ideology × age interaction ($p = 0.04$).

For the federal environmental spending question, in ten survey waves before 2012 (combined $N = 11,567$), we did not find a significant interaction between political ideology and age (all $ps > 0.21$). The only exception is in 2000 ($n = 1,434$), when the political ideology-age interaction was significant ($p < 0.02$). In 2012, 2016 and 2020 (combined $N = 14,685$), younger Americans were significantly less politically polarized than older Americans (all $ps < 0.01$). Annual regression outputs are available in S14 Table.

For the environmental regulations versus business question, in five survey waves before 2012 (combined $N = 5,845$), younger Americans were not differentially polarized compared to older Americans (all $ps > 0.11$). Since 2012 (combined $N = 14,685$), younger Americans have been significantly less politically polarized than older Americans in support for environmental regulations at the expense of business interests (all $ps < 0.001$). Annual regression outputs are available in S15 Table.

Analyzing other survey questions individually yielded convergent results. ANES asked additional specific climate policy questions 20 times over 1982–2020, such as support for a clean air and water tax (ANES 1990 and 1992) and offshore drilling (ANES 2012; full list in

S2 Table). 18 of 20 ANES questions analyzed reinforced the general finding that younger Americans have been less politically polarized than older Americans from 2012 on, but not before. The two exceptions were for: 1) an ANES 1990 question about whether to strictly enforce pollution standards (significant political ideology × age interaction, $p < 0.05$); and 2) an ANES 2012 question about nuclear energy (no significant political ideology × age interaction, $p = 0.08$). Otherwise, 14 climate policy questions asked before 2012 did not exhibit a significant political ideology × age interaction (all $ps > 0.10$), while for four climate policy questions asked after 2012, younger Americans were significantly less polarized than older Americans (all $ps < 0.05$). Annual regression outputs are available in S5 and S16–S34 Tables.

### 3.3 Research finding 3: Younger versus older Americans are not differentially politically polarized in three non-climate-related domains

Fig 4 shows that in modeling three ANES time-series on public support for policies not relating to climate change (e.g. defense spending, public vs private health insurance, and welfare spending), we found that younger people have not been consistently more politically polarized than older people since 2012. In nine survey waves that asked all three questions, only 2012 and 2020 featured statistically significant political ideology × age interactions ($p < 0.01$ in 2012 and 2020; for other years, $ps > 0.05$). Additionally, the direction of the interaction differs between 2012 and 2020; in 2012, older people are less politically polarized than younger people, but in 2020, older people are more polarized than younger people.

Analyzing time-series for the individual policy questions also yielded patterns of age-based political polarization that differed from the climate policy items. For defense spending, the political ideology × age interaction was significant only in 1994, 2008, and 2020 ($ps < 0.05$). For private versus public health insurance, the political ideology × age interaction was significant only in 2008 and 2012 ($ps < 0.05$). For welfare spending, the political ideology × age interaction was significant only in 1982 and 2012 ($ps < 0.05$). These time-series were distinct from the climate policy time series, which showed that younger Americans have been consistently less politically polarized since 2012 but not before. Within-wave estimates of the political ideology × age interaction effect size for the non-climate survey items are available in S35–S37 Tables.

## 4. Discussion

In large-scale time-series analyses of public opinion data, we found that younger Americans were less politically polarized than older Americans when it comes to support for climate policies in 2020. Longitudinal analyses spanning 1982–2020 suggest that this pattern has been consistent since 2012, but that it was not consistently present in 2008 and years prior. Additionally, lesser political polarization in younger adults was not found for three other politically polarized policy domains, suggesting that differential age-based political polarization might be unique to Americans' climate policy attitudes.

Younger Americans' being less politically polarized compared to older Americans in climate policy support is notable because polarization is a barrier to adopting urgently needed climate solutions [19, 21]. Based on our analysis, environmentalists and policymakers trying to build broad appeal for climate reforms will find some support among younger conservatives, who are consistently closer to liberals' climate policy attitudes than are older conservatives. To appeal to older conservatives, existing research on legacy motivations suggests that reminding people about the future generations that will be impacted by climate change increases their likelihood of donating to a pro-environmental group and cooperating with others to achieve shared climate goals [56, 57]. We encourage future experimental studies on the comparative efficacy of such pro-climate appeals to younger versus older conservatives, especially those

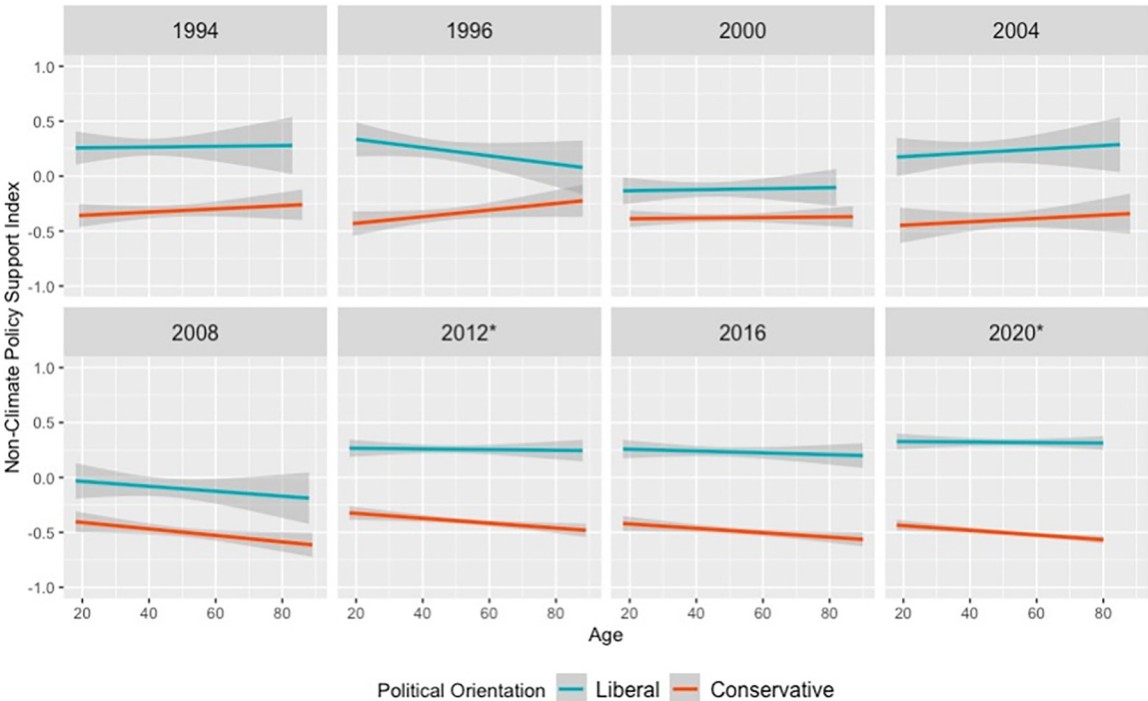

**Fig 4. Americans are not consistently differentially politically polarized based on age in support for defense spending, public health insurance, and welfare spending.** Note: Response variables are displayed with 95%-confidence interval smoothing. Years with an asterisk (e.g. 2012*) indicate significant political ideology × age interaction at $p < 0.05$, whereas non-asterisked survey years do not exhibit significant age-based political polarization. In all surveys, participants expressed their support for three non-climate policies, as well as their political ideology (1 = *extremely liberal* to 7 = *extremely conservative*) and age. Models treated political ideology as a continuous variable, but for visualization purposes, people who responded 1 or 2 are coded as "liberal", and people who responded 6 or 7 coded as "conservative". A version of this chart with 1992 non-climate data (excluded here for space considerations) and political moderates included is available in S2 Fig. For the federal spending on defense question, participants were asked, "Some people believe that we should spend much less money for defense. Suppose these people are at one end of a scale, at point 1. Others feel that defense spending should be greatly increased. Suppose these people are at the other end, at point 7. And, of course, some other people have opinions somewhere in between, at points 2, 3, 4, 5 or 6. Where would you place yourself on this scale, or haven't you thought much about this?" Responses were reverse-scored such that values ranged from options ranged from 1 = *greatly increase defense spending* to 7 = *greatly decrease defense spending*. For the public vs private health insurance question, participants were asked, "There is much concern about the rapid rise in medical and hospital costs. Some people feel there should be a government insurance plan which would cover all medical and hospital expenses for everyone. Suppose these people are at one end of a scale, at point 1. Others feel that all medical expenses should be paid by individuals through private insurance plans like Blue Cross or other company paid plans. Suppose these people are at the other end, at point 7. And, of course, some other people have opinions somewhere in between, at points 2, 3, 4, 5, or 6. Where would you place yourself on this scale, or haven't you thought much about this?" Responses were reversed-scored such that they ranged from 1 = *private insurance plan* to 7 = *government insurance plan*. For the welfare spending question, participants were asked, "Should federal spending on welfare programs be increased, decreased, or kept the same?", with responses coded such that 1 indicated a preference to *increase welfare spending*, while 0 reflected *keep welfare spending the same* or *decrease welfare spending*.

with research designs that account for political partisans' culturally constrained worldviews and the biasing effects of motivated reasoning [22, 26].

One possible explanation for lesser political polarization among younger Americans since 2012 is that today's cohort of young people, including young conservatives, is more aware of threats posed by climate change than older generations [34]. Such *cohort effects* could be due to the rising prominence of environmental activists with large youth social media followings, like Greta Thunberg [37]. Youth-specific motivation to address climate change might also be caused by feelings of climate-related anger and guilt, which increased in Millennial and Generation Z Americans, but not other age groups, over 2010–2019 [35]. Youth-specific motivations might also be attributable to today's media environment, which prominently discusses climate

change, affords social capital to environmental activists and helps foster pro-climate social norms [36, 38]. Testing for cohort effects would require analyzing a dataset that repeatedly studied these mechanisms in interviews with the same sample over time, whereas the ANES interviews new cross-sections of Americans with every wave.

An additional possibility could be that older people's climate change views are informed by political ideology to a greater degree than are younger people's due to effects of aging on decision making. Psychological studies of aging find that older adults are more likely than younger adults to avoid making decisions that elicit negative emotions, which might involve dismissing thoughts about climate policies [40, 42]. Additionally, older adults might seek to simplify their choices by considering fewer choice options [58], perhaps due to lower fluid intelligence as a result of aging [43]. Either mechanism could explain why older people might prefer to avoid deep reflection about climate change, and instead form climate policy opinions in accordance with their political ideology. That said, there is only limited evidence that older adults preferentially rely on political ideology, choice constraint or other decision-making shortcuts [45], while other authors do not find age-related differences [59].

Overall, younger Americans have been less politically polarized than older Americans since 2012 when it comes to support for policies that address climate change. We demonstrated this with large, nationally representative samples, showed consistency in climate attitudes across a plethora of specific climate policies, and drew contrasts to three other policy topics that did not feature clear differential political polarization based on age. Taken together, this robust tendency can inform policy debates around climate change and offer hope to those who seek to overcome partisan divisions to implement urgently needed climate solutions.

## Supporting information

**S1 Appendix. Complete question wording and coding decisions.**
(DOCX)

**S2 Appendix. Modeling details.**
(DOCX)

**S1 Fig. Younger Americans have been less politically polarized than older Americans on climate policy support since 2012, but not before (Except 1990).** Note: Response variables are displayed with 95%-confidence interval smoothing. Years with an asterisk (e.g. 2012*) indicate significant political ideology × age interaction at $p < 0.05$, whereas non-asterisked survey years do not exhibit significant age-based political polarization. In all surveys, participants expressed their support for climate policies, as well as their political ideology (1 = *extremely liberal* to 7 = *extremely conservative*) and age. Models treated political ideology as a continuous variable, but for visualization purposes, people who responded 1 or 2 are coded as "liberal", 3 or 4 or 5 are coded as "moderate", and people who responded 6 or 7 coded as "conservative". A listing of questions that were indexed to create each year's climate policy support index is available in the following tables for each modeling year.
(DOCX)

**S2 Fig. Americans are not differentially politically polarized based on age in support for defense spending, public health insurance, and welfare spending (Moderates included).** Note: Response variables are displayed with 95%-confidence interval smoothing. Years with an asterisk (e.g. 1994*) indicate significant age-based political polarization at the 0.05-significance level, whereas non-asterisked survey years do not exhibit significant age-based political polarization. In all surveys, participants expressed their support for three non-climate policies, as well as their political ideology (1 = *extremely liberal* to 7 = *extremely conservative*) and age.

Models treat political ideology as a continuous variable, but for visualization purposes, people who responded 1 or 2 are coded as "liberal", people who responded 6 or 7 coded as "conservative", and all others as "moderate".
(DOCX)

**S1 Table. American national election studies annual sample sizes, crosstab by political ideology and age group.** In all surveys, participants expressed their support for a range of climate policies, as well as their political ideology (1 = extremely liberal to 7 = extremely conservative) and age. Models treated political ideology as a continuous variable, but for tabulation purposes, people who responded 1 or 2 are coded as "liberal", and people who responded 6 or 7 coded as "conservative". For age, respondents are grouped into either "younger" (under 40 years old), "middle-aged" (40–60 years old), and "older" (over 60 years old). "% ANES Sample Retained" is the percentage of the respondents in the original ANES datafiles that we were able to model, with the remainder being excluded due to missing data.
(DOCX)

**S2 Table. ANES Environment and climate policy questions and response scales (Non-time-series items).**
(DOCX)

**S3 Table. Regression model for federal spending on the environment survey question (ANES 2020; logistic regression).**
(DOCX)

**S4 Table. Regression model for environment regulations vs business interests survey question (ANES 2020; linear regression).**
(DOCX)

**S5 Table. Regression model for regulations on greenhouse gas emitters survey question (ANES 2020; linear regression).**
(DOCX)

**S6 Table. Regression model for climate policy support index in ANES 1990 (three-item index; linear regression).**
(DOCX)

**S7 Table. Regression model for climate policy support index in ANES 1992 (four-item index; linear regression).**
(DOCX)

**S8 Table. Regression model for climate policy support index in ANES 1996 (ten-item index; linear regression).**
(DOCX)

**S9 Table. Regression model for climate policy support index in ANES 2000 (three-item index; linear regression).**
(DOCX)

**S10 Table. Regression model for climate policy support index in ANES 2008 (five-item index; linear regression).**
(DOCX)

**S11 Table. Regression model for climate policy support index in ANES 2012 (four-item index; linear regression).**
(DOCX)

**S12 Table. Regression model for climate policy support index in ANES 2016 (four-item index; linear regression).**
(DOCX)

**S13 Table. Regression model for climate policy support index in ANES 2020 (three-item index; linear regression).**
(DOCX)

**S14 Table. Annual regression models for federal spending on the environment ANES time-series (logistic regressions).**
(DOCX)

**S15 Table. Annual regression models for environmental regulations versus business interests ANES time-series (linear regressions).**
(DOCX)

**S16 Table. Regression model for clean air & water tax survey question (ANES 1990; logistic regression).**
(DOCX)

**S17 Table. Regression model for enforcing strict pollution regulations survey question (ANES 1990; logistic regression).**
(DOCX)

**S18 Table. Regression model for clean air & water tax survey question (ANES 1992; logistic regression).**
(DOCX)

**S19 Table. Regression model for enforcing strict pollution regulations survey question (ANES 1992; logistic regression).**
(DOCX)

**S20 Table. Regression model for pollution cleanup as a foreign policy goal survey question (ANES 1992; logistic regression).**
(DOCX)

**S21 Table. Regression model for improving and protecting the environment survey question (ANES 1996; logistic regression).**
(DOCX)

**S22 Table. Regression model for reducing air pollution survey question (ANES 1996; logistic regression).**
(DOCX)

**S23 Table. Regression model for managing natural resources survey question (ANES 1996; logistic regression).**
(DOCX)

**S24 Table. Regression model for cleaning up lakes and parks survey question (ANES 1996; logistic regression).**
(DOCX)

**S25 Table. Regression model for cleaning up hazardous or toxic waste survey question (ANES 1996; logistic regression).**
(DOCX)

**S26 Table. Regression model for reducing solid waste and garbage survey question (ANES 1996; logistic regression).**
(DOCX)

**S27 Table. Regression model for addressing global warming survey question (ANES 1996; logistic regression).**
(DOCX)

**S28 Table. Regression model for fuel standards survey question (ANES 2008; linear regression).**
(DOCX)

**S29 Table. Regression model for power plant emission standards survey question (ANES 2008; linear regression).**
(DOCX)

**S30 Table. Regression model for gasoline tax survey question (ANES 2008; linear regression).**
(DOCX)

**S31 Table. Regression model for nuclear power plants survey question (ANES 2012; logistic regression).**
(DOCX)

**S32 Table. Regression model for offshore drilling survey question (ANES 2012; logistic regression).**
(DOCX)

**S33 Table. Regression model for fracking survey question (ANES 2016; logistic regression).**
(DOCX)

**S34 Table. Regression model for federal action on rising temperatures survey question (ANES 2016; linear regression).**
(DOCX)

**S35 Table. Annual regression models for federal spending on defense ANES time-series (linear regressions).**
(DOCX)

**S36 Table. Annual regression models for public versus private health insurance ANES time-series (linear regressions).**
(DOCX)

**S37 Table. Annual regression models for federal spending on welfare ANES time-series (logistic regressions).**
(DOCX)

## Acknowledgments

We thank Norbert Schwarz and his SEEP lab members, as well as participants of the Social Psychology brown bag series at the University of Southern California, for their helpful feedback.

## Author Contributions

**Conceptualization:** Joshua F. Inwald, Christopher D. Petsko.

**Data curation:** Joshua F. Inwald, Wändi Bruine de Bruin.

**Formal analysis:** Joshua F. Inwald.

**Funding acquisition:** Christopher D. Petsko.

**Methodology:** Joshua F. Inwald, Wändi Bruine de Bruin, Christopher D. Petsko.

**Project administration:** Joshua F. Inwald.

**Software:** Joshua F. Inwald.

**Visualization:** Joshua F. Inwald.

**Writing – original draft:** Joshua F. Inwald.

**Writing – review & editing:** Joshua F. Inwald, Wändi Bruine de Bruin, Christopher D. Petsko.

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
