## [Decision Letter · Decision Letter 0]

5 Jan 2024

PONE-D-23-36855Younger Americans are less politically polarized than older Americans about climate policies (but not about other policy domains)PLOS ONE

Dear Dr. Inwald,

Thank you for submitting your manuscript to PLOS ONE. After careful consideration, we feel that it has merit but does not fully meet PLOS ONE’s publication criteria as it currently stands. Therefore, we invite you to submit a revised version of the manuscript that addresses the points raised during the review process.

We look forward to receiving your revised manuscript.

Kind regards,

José Gutiérrez-Pérez

Academic Editor

PLOS ONE

Journal Requirements:

Reviewers' comments:

Reviewer's Responses to Questions

**Comments to the Author**

1. Is the manuscript technically sound, and do the data support the conclusions?

Reviewer #1: Partly

Reviewer #2: Yes

2. Has the statistical analysis been performed appropriately and rigorously? 

Reviewer #1: Yes

Reviewer #2: Yes

3. Have the authors made all data underlying the findings in their manuscript fully available?

Reviewer #1: Yes

Reviewer #2: Yes

4. Is the manuscript presented in an intelligible fashion and written in standard English?

Reviewer #1: Yes

Reviewer #2: Yes

5. Review Comments to the Author

Reviewer #1: Dear authors,

Thank you for sharing your work. It highlights the significant statistical effort put forth and the interest and opportunity presented by conducting a longitudinal study of secondary analysis, as outlined in your manuscript.

The manuscript begins its introduction by establishing the need for measures of adaptation and mitigation to climate change, referencing the IPCC and WHO. It then articulates the slow response of the United States in relation to the Paris Agreement, linking it to the imperative of promoting public awareness and the current situation in the United States. The narrative continues by exploring the influence of political polarization on attitudes and beliefs regarding anthropogenic climate change, focusing on the differences between young and adult populations. The justification for the study's relevance lies in the national context and the absence of studies addressing political polarization, not only in climate policies but also in other political dimensions.

The text proceeds to describe the study, stating its main objective: to understand the relationships between ideology and age in supporting climate policies. Additionally, it aims to contextualize previous studies on polarization in climate policies by analyzing attitudes toward various climate policies, examining surveys conducted by ANES over the last four decades, and finally, comparing different attitudes between climate policies and other political domains. The manuscript concludes by posing three research questions.

The subsequent section, "Materials and Methods," details the public opinion surveys conducted by ANES, serving as valuable secondary data for analysis. It covers participants, the application process, variables to be analyzed, and the coding procedures. The presentation of results highlights three key findings: that in 2020, young Americans are less politically polarized than their adult counterparts; that young Americans have polarized less than their adult counterparts in relation to climate policies during the 2008-2012 period; and that both young and adult populations show no differences in political polarization regarding political domains unrelated to climate change.

Finally, the text concludes with a discussion section.

During the initial reading of the manuscript, I found myself a bit disoriented due to the utilization of data from 16 surveys spanning 38 years (1982-2020), with variations in some of the questions across different waves. It's important to note that such situations are inherent to this type of opinion surveys, which draw on national demography data that often alter their formulations from one wave to another.

Upon subsequent readings, certain doubts and concerns have arisen that I believe need to be addressed before recommending the publication of your work. Below, I outline the reasons behind this decision. I trust that you will receive my observations and comments with the sole purpose of facilitating improvements for the successful publication of your work.

Introduction

I find the introduction somewhat limited and burdened with references that could or should be reduced, as they do not contribute additional information and relevance to the argument. For instance, references 8, 9, and 10 pertain to three documents on the same 2019 survey (World Risk Poll), and references 14, 15, and 16, two of which are studies from the Yale Program on Climate Change, widely known and referencing one of them would likely suffice.

Furthermore, these references are not utilized to compare and discuss the results in the discussion section. However, new references are introduced in the discussion section to analyze the findings. In my experience in reading, writing, or reviewing articles, as well as in various peer review training courses and seminars I have attended, it is customary for the introduction and the state of the art to present the literature guiding and contextualizing the study. Subsequently, the results are discussed in relation to the previously mentioned literature. This structure is not evident in this work. Out of the 27 citations in this introductory section, only 2 (citations 21 and 23) are used to discuss the study's results.

I believe there is a need for a more in-depth literature review that positions the reader in the current state of research, particularly regarding ideological polarization among different age groups concerning climate policies and other political dimensions unrelated to climate change. To achieve this, you can make use of references 39 to 48, which you have employed to discuss your results. Additionally, I recommend exploring studies by Professor Dan Kahan of the Cultural Cognition Project (http://culturalcognition.squarespace.com/ ), where you will find research on political polarization, worldviews, and perceptions on climate and other controversial topics (weapons, nuclear energy, etc.).

Materials and Methods

I suggest restructuring this section to enhance clarity and provide a transparent presentation of the followed methodological process.

The section begins by describing the work of the American National Election Studies (ANES), which is logical and coherent as the source of secondary data for the study. However, it would be beneficial to include information on where these data can be accessed. The current description is limited to a 6-line paragraph, and subsequent sub-sections continue to provide details about ANES that I believe should be incorporated at the beginning of this section, such as the paragraph between lines 127 and 130. Moreover, in my opinion, the information provided in Table 1 may not be highly relevant and could be included in supplementary materials. The footnote information could be incorporated into the description of the ANES surveys at the start of the section. If deemed necessary by the authors, they can briefly refer to the relevant information from the table. Additionally, it would be useful to include a sub-section (2.1) with a title for this ANES description.

In sub-section 2.2, Measures, I would take the opportunity to include a table describing the variables of interest, given the removal of Table 1. This suggestion aims to allow readers to easily refer back and identify the variables discussed throughout the article. For example:

Variable Wording Scale

Independent variables Liberal-Conservative Political Ideology Where would you place yourself on this scale, or haven’t you thought much about this? 7-point scale…

Age Birth year, month and day

Climate policy time-series items Federal Spending on the environment

Environment regulations…

Non-climate time-series items Federal Spending…

Private versus…

Federal Spending on…

Regarding Table 2, which displays other questions about climate policy, I don't believe many of these items are directly related to climate policies; rather, they pertain more to environmental policies. In my opinion, the first 9 items do not align with climate policies but instead focus on environmental, well-being, and social health considerations. It is from the item "Address Global warming" to "Regulations on GHG emitters" that the questions become directly associated with climate policies.

Finally, I fail to see the relevance of reference (31). I suggest considering the possibility of removing it.

Results

In this section, I also have some questions. Shouldn't the strategic analysis be included in the methods section? This may be a matter of personal opinion or preferences, but upon reaching this section, I expected to start reading about results. However, the explanation of methodological aspects of the study continues. Don't you think this breaks the flow?

Moreover, you previously explained the variables of interest in the study. Nevertheless, in this section, you revisit explanations about these variables and others like educational level, which are ultimately not included in the model. I believe the content of this section up to sub-section 3.2 should be incorporated into the methods section, please consider if it's appropriate. Additionally, I think the R programming language and its various packages are widely known and may not require references for support and justification (references 36 and 37). Furthermore, consider whether it would be interesting to present a diagram where readers can observe the regression models and how variables respond to these models.

Regarding the rest of the section, as I mentioned earlier, I find it a bit overwhelming due to the number of variables used. Having the suggested table, which systematically describes the variables used, might make reading the results easier. In any case, I encourage you to attempt restructuring this section so that you are clearly answering the research questions. I believe this could enhance the understanding of this section and its results.

Discussion

As I mentioned earlier, I believe that for this discussion section to have coherence and meaning, it would be necessary to establish a prior state of the art, anticipating the reader's understanding of the study's context and allowing the conclusion of the study with this final discussion section.

I hope you find my observations constructive. I consider that you have done a great job, and your results are valuable for the field of communication and climate policies. However, I believe there is room for improvement in presenting your study to reach a broader audience.

Kind regards.

Reviewer #2: This paper uses American National Election Studies data to examine trends in polarization of American attitudes regarding policies on climate change and other issues. The authors find that, starting in 2012, younger Americans have been less polarized along partisan lines than older Americans, and that this age-based distinction in the extent of polarization does not exist on other important policy issues. In the conclusion of the article, the authors also thoughtfully consider possible explanations of the difference they find between younger and older Americans on climate change. The paper's statistical analysis is technically sound and the paper is very well written.

6. PLOS authors have the option to publish the peer review history of their article (what does this mean?). If published, this will include your full peer review and any attached files.

Reviewer #1: No

Reviewer #2: **Yes: **Jordan Tama

---

## [Author Response · Author response to Decision Letter 0]

19 Mar 2024

Please see response to reviewer comments in the uploaded Word doc

---

## [Decision Letter · Decision Letter 1]

4 Apr 2024

Younger Americans are less politically polarized than older Americans about climate policies (but not about other policy domains)

PONE-D-23-36855R1

Dear Dr. Inwald,

We’re pleased to inform you that your manuscript has been judged scientifically suitable for publication and will be formally accepted for publication once it meets all outstanding technical requirements.

Kind regards,

José Gutiérrez-Pérez

Academic Editor

PLOS ONE

Reviewer Comments #1: 

Dear authors,

Thank you for addressing my concerns. I believe the revised manuscript has significantly enhanced its clarity. I find your work to be highly intriguing for climate change communication and education, not only within the United States but also internationally.

Best regards,

---

## [Editor Report · Acceptance letter]

26 Apr 2024

PONE-D-23-36855R1 

PLOS ONE

Dear Dr. Inwald, 

I'm pleased to inform you that your manuscript has been deemed suitable for publication in PLOS ONE. Congratulations! Your manuscript is now being handed over to our production team.

Kind regards, 

on behalf of

Dr. José Gutiérrez-Pérez 

Academic Editor

PLOS ONE